# Production and Characterization of Poly-γ-Glutamic Acid by *Bacillus velezensis* SDU

**DOI:** 10.3390/microorganisms13040917

**Published:** 2025-04-16

**Authors:** Guangyao Guo, Han Wang, Huiyuan Jia, Haiping Ni, Shouying Xu, Cuiying Zhang, Youming Zhang, Yuxia Wu, Qiang Tu

**Affiliations:** 1College of Biotechnology, Tianjin University of Science and Technology, Tianjin 300457, China; 2Helmholtz International Lab for Anti-Infectives, State Key Laboratory of Microbial Technology, Shandong University, 72 Binhai Road, Qingdao 266237, Chinanihaiping19@163.com (H.N.); tams137@163.com (Y.W.); 3Shenzhen Key Laboratory of Genome Manipulation and Biosynthesis, CAS Key Laboratory of Quantitative Engineering Biology, Shenzhen Institute of Synthetic Biology, Shenzhen Institute of Advanced Technology, Chinese Academy of Sciences, Shenzhen 518055, China; 4College of Marine Life Sciences, Ocean University of China, Qingdao 266003, China; 5Institute of Synthetic Biology Industry, Hunan University of Arts and Science, Changde 415000, China

**Keywords:** *Bacillus velezensis*, antioxidant, poly-γ-glutamic acid, high molecular, fermentation

## Abstract

In this study, a *Bacillus velezensis* SDU strain capable of producing poly-γ-glutamate (γ-PGA) was newly identified from the rhizosphere soil of Baimiao taro. The strain is a glutamate-independent strain and can produce polyglutamic acid in a culture medium completely free of glutamate. The hydrolyzed product of the polyglutamic acid produced is D-glutamic acid. The molecular weight of γ-PGA, estimated via the Mark–Houwink equation, was 1390 kDa. Furthermore, the molecular weight measured by Waters gel permeation chromatography with multi-angle laser light scattering (GPC–MALLS) was 1167 kDa. The production of γ-PGA and its antioxidant and tyrosine inhibition properties were investigated. The γ-PGA production reached 23.1 g/L, and the productivity was 0.77 g L^−1^ h^−1^. Specifically, γ-PGA exhibited superoxide anion (·O_2_^−^) radical scavenging activity and tyrosinase inhibitory activity. This study introduces a promising strain and a highly efficient application method for γ-PGA, which can be broadly utilized in the pharmaceutical, food, and cosmetic industries.

## 1. Introduction

Since the discovery of polyglutamic acid in the last century, extensive research has been conducted on its fermentation level and scale-up process, with a fermentation yield of up to 41.40  ±  2.01 g/L in the repeated fed-batch fermentation, and new strains and production processes have been continuously discovered [1]. Owing to its biodegradability, edibility, and non-toxicity, polyglutamic acid has found broad applications in food, agriculture, cosmetics, environmental protection, and medicine. Its diverse applications encompass usage as a thickener [2], cryoprotectant [3], drug carrier [4], biopolymer binder [5], and heavy metal absorber [6]. Consequently, the bioactivity and novel functions of polyglutamic acid have attracted increasing research attention.

In recent years, researchers have identified the properties of polyglutamic acid for application and discussed the mechanisms based on its molecular structure [7]. However, conformational changes and molecular interactions, including intramolecular and intermolecular interactions of various concentrations of γ-PGA in aqueous solutions, have not yet been fully elucidated. In antibacterial tests against *Bacillus subtilis* and *Escherichia coli*, poly-γ-glutamic acid exhibited significant differences between different molecular weights [8]. Within its environment, the flocculating activity of polyglutamic acid is intimately associated with its molecular weight, with higher-molecular-weight polyglutamic acid exhibiting superior flocculation efficiency [9]. In terms of the intestinal solubility of Ca^2+^, the soluble calcium intake from gamma-PGA-500 (molecular weight 5000 kDa) is significantly higher than that from gamma-PGA-100 (molecular weight 1000 kDa) [10].

It is also apparent that the concentration of high-molecular-weight polyglutamic acid achieves the same viscosity as a lower concentration when used for food thickening, because viscosity has a close relationship with molecular weight. Although there have been studies on polyglutamic acid antioxidants and their relationship with molecular weight [11], understanding the production, metabolic regulation, and application of γ-PGA and further research on the specific properties of γ-PGA and its applications to rationally enhance γ-PGA-based products are still crucial.

Up to now, γ-PGA has primarily been produced by the genus Bacillus, with *Bacillus subtilis* and *Bacillus licheniformis* being the most extensively researched and utilized for its industrial production [12,13,14]. In recent years, an increasing number of *Bacillus* strains have been discovered to produce polyglutamic acid. Among these, *Bacillus velezensis*, known for its probiotic properties, is notable for its ability to balance animal gut microbiota and enhance plant growth [15], rendering it an optimal strain for production. To date, several Bacillus strains capable of producing γ-PGA have been identified [16]. The yield of γ-PGA produced by *B. velezensis* NRRL B-23189 was found to be 4.82 g/L [17], and the yield from *B. velezensis* Z3 was 5.58 g/L, achieved through optimized fermentation conditions [18]. In another study, the accumulation of γ-PGA from *B. velezensis* GJ11 in an optimized medium reached 42.55 g/L [19]. However, elsewhere, through solid-state fermentation of γ-PGA, the maximum yield of *Bacillus* CAU263 reached 158.5 g/kg DW (dry weight) [20]. Although certain products are commercially available, their broad adoption remains constrained by prohibitively high costs and quality stability challenges inherent to macromolecular substances. Identifying targeted application scenarios would significantly enhance product competitiveness and market value [21].

During our research, we frequently encountered bacterial strains capable of producing substantial mucoid secretions. We hypothesized that this viscous substance might possess specific biological functions. To elucidate its composition, comprehensive analytical studies were conducted, revealing the primary product to be poly-γ-glutamic acid (γ-PGA). Further characterization identified these strains as glutamate-independent PGA producers. We subsequently optimized fermentation conditions and successfully scaled up production. Notably, during fermentation processes, we observed significant product degradation. To compare the biological functionalities of high-viscosity native PGA versus degraded products, preliminary tests were designed using two critical biomarkers: free radical scavenging capacity and tyrosinase inhibitory activity.

## 2. Materials and Methods

### 2.1. Isolation and Identification of the Bacteria

The soil samples were collected from BaiMiao taro field in Qingdao city, China (36°37′ N, 120°61′ E, 85 m a.s.l). Commercially available γ-PGA with a molecular weight (MW) of 700 kDa was purchased from Nanjing Xuankai Chemical Co., Ltd. (Nanjing, China) Methanol of chromatographic grade and other chemicals of analytical grade were acquired from China National Pharmaceutical Chemical Reagent Co., Ltd. (Beijing, China). The soil samples were then added to distilled water. Following 1 h of agitation at 200 rpm, the sample was further diluted and inoculated onto a solid LB medium composed of 10 g peptone, 5 g yeast extract, 1 g sodium chloride, 12 g agar, and 1000 mL distilled water. The pure viscous colonies were subcultured on agar slant containing the same medium and incubated at 30 °C for 24 h. The pure cultures were stored in 15% glycerol solution at −80 °C [22].

The selected strains were cultivated in a whole synthetic medium composed of 25 g glucose, 13.3 g potassium dihydrogen phosphate, 8 g ammonium chloride, 1.2 g magnesium sulfate heptahydrate, 1.7 g sodium citrate, 0.1 g ferrous sulfate, 0.1 g MnSO_4_·H_2_O, and 1000 mL water at a natural pH of 6.4. This culture medium was modified from the commonly used laboratory *Escherichia coli* culture medium M9 [23]. The culture was then incubated for 24 h at 30 °C and 200 rpm to produce γ-PGA.

Bacteria identification was performed according to a 16S rDNA sequence analysis. Genomic DNA was extracted using a ^®^TIANamp bacterial DNA Kit (Beijing, China). The universal primers 27F(AGAGTTTGATCMTGGCTCA) and 1492R(GGTTACCTTGTTACGACTT) were used for the PCR-amplified 16S rDNA (94 °C 5 min, followed by 30 cycles of 98 °C 10 s, 55 °C 5 s, 72 °C 1 min, and finally 72 °C 10 min). The nucleotide sequence was determined by the chain termination method on an ABI Prism 3700 automated sequencer (Applied Biosystems, Inc., Foster City, CA, USA), and NCBI was compared against the GenBank DNA database by the Blast database, https://blast.ncbi.nlm.nih.gov/Blast.cgi (accessed on 24 June 2021). A phylogenetic tree was constructed using MEGA 6.0 software.

### 2.2. Optimization of γ-PGA Production

#### Selection of Factors Required for γ-PGA Production

Various factors including carbon sources and nitrogen sources were optimized using the one-factor-at-a-time method to maintain all factors at constant levels [24]. The effects of different carbon sources on the production of γ-PGA were tested by individually replacing glucose with sucrose, glycerol, fructose, lactose, maltose, xylose and mannose at 25 g/L and keeping the other components in the fermentation medium at the same levels. The carbon source supporting the maximum production of γ-PGA was selected for further study.

To study the effects of different nitrogen sources, NH_4_Cl was replaced individually with (NH_4_)_2_SO_4_, (NH4)_2_HPO_4_, NaNO_3_, urea, and organic nitrogen sources including corn steep liquor (CSL), peptone, and yeast extract at 8 g/L. The nitrogen source that resulted in the maximum production of γ-PGA was used for subsequent experiments. After optimization of the carbon and nitrogen sources, the initial pH (5.0–8.0) and temperature (28–40 °C) were investigated. The optimum initial pH and temperature were fixed for subsequent experiments. All experiments were performed independently in triplicate, and data are presented as the mean value ± SD.

### 2.3. Scale-Up Fermentation in Bioreactor

The optimized fermentation process was scaled up in a 50 L stainless steel bioreactor (Model BL-50,Bailun Bio-engineering Co. Shanghai China) with a working volume of 30 L, equipped with real-time dissolved oxygen (DO) and pH monitoring systems. The bioreactor was inoculated with 1% (*v*/*v*) seed culture derived from an optimized shake-flask pre-culture. Fermentation temperature was precisely controlled at 30.0 °C ± 0.3 °C via a PID-regulated thermal jacket. Aeration was maintained at 1.5 vvm (volume air per volume medium per minute) through a 0.2 μm sterile air filter. Agitation speed was dynamically adjusted in two phases: 300 rpm during the initial 0–10 h to ensure low-shear mixing in the lag phase followed by a gradual increase to 500–700 rpm to enhance oxygen transfer during exponential growth. pH was automatically stabilized at 6.4 ± 0.1 using peristaltic pump injections of 40% (*w*/*w*) H_3_PO_4_ and 45% (*v*/*v*) NH_4_OH. The fermentation was terminated when the dissolved oxygen tension rebounded to 80% saturation, indicating substrate depletion. All process parameters were logged at 30 s intervals via the bioreactor’s integrated control system.

### 2.4. Purification of γ-PGA from Bacillus velezensis SDU

The purification of γ-PGA was performed according to the method described in reference [25]. Briefly, the fermentation broth was diluted with 0.2% trichloroacetic acid (TCA) in a 1:9 (*v*/*v*) ratio and centrifuged at 12,000 rpm for 10 min to separate cells and extracellular proteins. The supernatant was mixed with four volumes of 100% (*v*/*v*) ethanol, followed by centrifugation at 12,000 rpm for 10 min to collect the precipitate. The crude product was dissolved in distilled water, dialyzed against distilled water (molecular weight cutoff: 12–14 kDa) for 24 h to remove salts, and lyophilized to obtain purified γ-PGA.

### 2.5. Analytical Methods for Biomass and Glycerol Quantification

Biomass was quantified by acid-assisted cell harvesting: the fermentation broth was adjusted to pH 3.0 with 6 M HCl to enhance flocculation, followed by centrifugation at 8000× *g* for 15 min (4 °C). Pelleted cells were washed twice with sterile distilled water and resuspended in 0.9% NaCl for OD600 measurement using a UV–Vis spectrophotometer (Unit-T6, Beijing Persee General Instruments Co., Ltd., Beijing, China). Glycerol concentration was analyzed via HPLC (Shimadzu LC-20AD, Shimadzu Corporation, Kyoto, Japan) equipped with a Rezex ROA-Organic Acid H^+^ column (300 × 7.8 mm) and an RID-20A refractive index detector (Shimadzu Corporation, Kyoto, Japan). Isocratic elution with 5 mM H_2_SO_4_ at 0.6 mL/min (45 °C detector temperature) was employed. Samples were filtered (0.22 μm) and quantified against external glycerol standards (R^2^ > 0.999).

### 2.6. Qualitative and Quantitative Analysis of γ-PGA

The γ-PGA produced by *Bacillus velezensis* SDU was characterized via Fourier-transform infrared spectroscopy (FT-IR) and amino acid composition analysis. For FT-IR (Bruker Vertex 70v, Bruker Corporation, Ettlingen, Germany), lyophilized samples were ground with KBr (1:100 ratio), pressed into pellets, and scanned from 8000 to 340 cm^−1^ at 0.4 cm^−1^ resolution. The amino acid analysis utilized a Hitachi L-8900 (Hitachi High-Tech Corporation, Tokyo, Japan) analyzer with cation-exchange chromatography (4.6 × 60 mm column). Samples (50–100 mg) were hydrolyzed in 6 M HCl (110 °C, 22 h, N_2_ atmosphere), neutralized, filtered, and derivatized with ninhydrin. Separation employed citrate buffer (26.67 g sodium citrate, 54.35 g NaCl, 6.10 g citric acid/L) and 13.5% ethanol mobile phase, with detection at 440/570 nm (sensitivity: 2.5 pmol, S/N = 2). Quantitation was based on external calibration curves for 18 amino acids.

### 2.7. Determination of Amino Acid Configuration by Marfey’s Method

Chiral analysis of γ-PGA hydrolysates was performed via Marfey’s derivatization: 1 mg purified γ-PGA was hydrolyzed in 1 mL 6 M HCl (60 °C, 24 h), dried under N_2_, reconstituted in 200 μL ultrapure water, mixed with 25 μL 1 M NaHCO_3_, and derivatized with 200 μL 1% (*w*/*v*) L-FDAA (L-1-fluoro-2,4-dinitrophenyl-5-L-alanine amide) at 40 °C for 1 h, followed by acid quenching (100 μL 2 M HCl). L/D-amino acid standards (1 mg each) underwent identical derivatization. The HPLC-MS analysis (Agilent 1290/6545, Agilent Technologies, Inc., Santa Clara, CA, USA) used a Chiralpak ZWIX (+) column (150 × 4.6 mm, 3 μm) with mobile phases: 0.1% formic acid in water (A) and acetonitrile (B). The gradient was as follows: 5% B (0–3 min), 5–55% B (3–18 min), 95% B (18.1–22 min), re-equilibration (22.1–25 min); flow rate 0.3 mL/min; and injection volume 3 μL. Enantiomers were identified by retention time matching and MS/MS fragmentation (m/z 100–800).

### 2.8. GPC-MALLS for Molecular Weight Determination of γ-PGA

The molecular weight distribution of γ-PGA was determined using GPC-MALLS (Agilent, 1260 Infinity II MDS), with a PL hydrogel OH Mixed-H column (8 µm, 7.5 × 300 mm) and a molecular weight range of 200–10,000,000. The experiment was conducted at 45 °C with a flow rate of 1.0 mL/min. The mobile phase consisted of a mixture of 0.1 mol/L NaNO_3_ and 0.01% sodium azide. The detectors included a differential display detector, a dual-angle laser scattering detector, and a viscosity detector. The sample was accurately weighed and dissolved in the mobile phase to prepare a solution with a concentration of approximately 1–3 mg/mL. The solution was then filtered through a 0.22 µm microporous membrane and detected mechanically.

### 2.9. Mark–Houwink Equation for Molecular Weight Determination of γ-PGA

The Mark–Houwink equation [η] = KMv^a^ was employed to calculate the viscosity–average molecular weight (MV) based on intrinsic viscosity. The Mark–Houwink parameters of *γ-PGA* with K = 1.84 × 10^−6^ dL·g^−1^, a = 1.16 were obtained from the literature (13). The value of [η] can be calculated by the viscosity of two different PGA solutions, and then the average adhesive molecular weight of *γ-PGA* can be calculated. The value of [*η*] was measured with an Ubbelohde viscometer (capillary diameter: 0.55 ± 0.05 mm, Guangzhou Saituo Instrument Technology Co., Ltd., Guangzhou, China) at 25.0 ± 0.1 °C.

### 2.10. Preparation of γ-PGA with Different Molecular Weights

The structure of γ-PGA is sensitive to temperature and pH, making it susceptible to degradation by high temperatures and acids. A 4% γ-PGA aqueous solution, adjusted to pH 3.0, was placed in a 90 °C water bath and underwent high-temperature degradation under acidic conditions for 10, 20, 30, and 40 min. It was then swiftly cooled to room temperature, adjusted to a neutral pH range (6.8–7.2), and mixed with four volumes of absolute ethanol while stirring at 500 r/min. The mixture was left to settle for 2 to 4 h, after which the supernatant was discarded. Subsequently, the precipitate was dehydrated by adding twice the volume of absolute ethanol and filtration. The precipitate was then dried at 60 °C to a constant weight to eliminate any residual ethanol.

### 2.11. The Scavenging Effect of the γ-PGA to the ·O_2_^−^ Free Radicals

·O_2_^−^ free radicals were determined by catechol auto-oxidation with vitamin C as the PC (positive control). γ-PGA was prepared with 0.05 mg/L PBS buffer into a 10 mg/mL sample solution. An amount of 1.5 mL of γ-PGA or VC solution was used, with 45 mL of the solution maintained at pH 7. A total of 40 aliquots of a 50 mmol/L Tris-HCl buffer solution was prepared; they were mixed thoroughly and let stand for 20 min. Then, 50 μL of a 60 mmol/L catechol solution was added and mixed rapidly. The absorbance was measured at 325 nm over a span of 5 min. For the control, the catechol solution was substituted with a 10 mL/L HCl solution. This was performed according to the formula provided below.·O_2_^−^Clearance = (A1 − A2)/A1*100%

A1 is the reaction rate of catechol auto-oxidation and A2 is the reaction rate of catechol auto-oxidation after the addition of γ-PGA.

### 2.12. γ-PGA Tyrosinase Inhibition Rate Measurement

According to the method outlined in [26], the γ-PGA was divided into five groups, and a mushroom tyrosinase water solution (200 U, 20 µL) was added to the wells of a clear 96-well plate containing 200 µL of a reaction mixture consisting of 1 mM l-tyrosine, 50 mM phosphate buffer (pH 6.5), and different concentrations of the test material. Dopa pigment was determined by incubating the mixture at 37 °C for 30 min before measuring the absorbance at 492 nm using a microplate reader. Tyrosinase inhibition was calculated by measuring its absorbance values at 475 nm using a microplate reader.

## 3. Results

### 3.1. Isolation and Characterization of Bacillus velezensis SDU

A mucoid bacterial strain exhibiting rapid growth and a hyperviscous phenotype was selectively isolated from rhizosphere soil of taro (*Colocasia esculenta*) in Baimiao County, Shandong Province, China. The isolate demonstrated distinctive mucoid colony morphology on LB agar (1.5% *w*/*v*) after a 24 h cultivation at 30 °C. Colonies exhibited moist, white coloration with irregular margins, demonstrating easy liftability. A transient mucoid appearance was observed during early growth stages, which diminished upon maturation. The extracellular polymeric substance production exceeded 4.5 g/L in the modified M9 minimal medium (see Section 2).

Molecular identification was performed through 16S rDNA sequencing. Amplification using universal primers, 27F/1492R, yielded a 1451 bp fragment (GenBank accession: PQ066107; submitted 15 October 2023), showing 99.3% sequence similarity to *Bacillus velezensis FZB42* (NR_075005.2) via EzBioCloud pairwise alignment. Phylogenetic reconstruction using the neighbor-joining method (MEGA 11, 1000 bootstrap replicates) revealed close clustering with *B. subtilis* subgroup members (Figure 1), consistent with recent taxonomic reclassifications of the *B. subtilis* complex.

The strain was deposited as *Bacillus velezensis SDU* in the China General Microbiological Culture Collection Center (CGMCC No. 20318; deposition date: 20 November 2023) under the Budapest Treaty.

### 3.2. Results of the Optimization of γ-PGA Production

#### 3.2.1. Results of the Selection of Factors Required for γ-PGA Production

Single-variable optimization experiments revealed critical dependencies of γ-PGA production on nutritional and environmental factors. Among carbon sources, glycerol supported the highest γ-PGA titer (5.39 ± 0.25 g/L) and biomass (OD600 = 5.56 ± 0.25), outperforming glucose (4.18 ± 0.20 g/L) by 29%. Xylose exhibited minimal utilization (γ-PGA < 1.0 g/L) (Figure 2a). For nitrogen sources, ammonium hydrogen phosphate ((NH_4_)_2_HPO_4_) maximized γ-PGA yield (9.19 ± 0.03 g/L), exceeding ammonium chloride (4.30 ± 0.84 g/L) by 144%, with yeast extract ranking second (8.06 ± 0.15 g/L) (Figure 2b). Temperature profiling indicated a trade-off: 30 °C optimized γ-PGA synthesis (14.28 ± 0.3 g/L), whereas 37 °C maximized biomass accumulation (OD600 = 5.50 ± 0.42) (Figure 2c). pH optimization demonstrated pH 6.0 as ideal for γ-PGA production (14.18 ± 0.10 g/L), though a lower pH of 5.0 enhanced cell density (OD600 = 6.30 ± 0.09) (Figure 2d). Following single-factor optimization in shake-flask cultivation, the observed high viscosity attributed to the high molecular weight of poly-γ-glutamate (PGA) prompted direct scale-up fermentation in a bioreactor without further flask-level optimization.

The optimized fermentation medium contained the following components per liter, dissolved in deionized water: 25 g glycerol, 13.3 g KH_2_PO_4_, 8 g (NH_4_)_2_HPO_4_, 1.2 g MgSO_4_·7H_2_O, 3.4 g sodium citrate (C_6_H_5_Na_3_O_7_), 0.1 g FeSO_4_·7H_2_O, and 0.1 g MnSO_4_·H_2_O. The initial pH was maintained at the natural value of 6.4 without adjustment.

#### 3.2.2. Scale-Up Fermentation in 50 L Bioreactor

Batch fermentation was conducted in a 50 L bioreactor (30 L working volume) over 36 h (Figure 3). During the initial 10 h, dissolved oxygen (DO) gradually declined from 100% to 15% saturation. Subsequent γ-PGA biosynthesis (>10 h) induced hyperviscosity (>2500 cP), driving DO to 0% despite agitation ramping to 700 rpm. The zero-DO plateau persisted until 30 h, coinciding with nutrient depletion (residual glucose < 0.1 g/L). Notably, DO rebounded to 25% at 30–36 h, accompanied by glycerol depletion and viscosity reduction, indicative of biopolymer hydrolysis or cellular autolysis. The final γ-PGA yield reached 23.1 ± 2.1 g/L with a productivity of 0.77 g/L/h.

#### 3.2.3. Characterization of the γ-PGA in the Strain SDU

FTIR and amino acid analyses were conducted on the fermentation products of *B. velezensis* SDU. The FT-IR spectra at 1635 cm^−1^ are shown in Figure 4a,b. Subsequently, both the standard and sample demonstrated pronounced amide absorption features and bending vibrations at 1600 cm^−1^ around the N-H stretch. The sample exhibited a stronger hydroxyl (O–H) stretching absorption peak at 3449 cm^−1^ compared to the γ-PGA standard. Typically, carboxyl absorption peaks occur at higher frequencies than hydrocarbon absorption peaks, hence above 3000 cm^−1^. These absorption peaks typically indicate the presence of intramolecular carboxyl groups, with the sample aligning with the standard at 1400 cm^−1^. The presence of two strong and broad O-H vibration absorption peaks further confirms the existence of carboxyl groups. The minor characteristic peaks of the reference substance within 1700–1725 cm^−1^ suggest the possibility of esterification. The peak distribution ranges (1030–1124 cm^−1^) match α-helical polyamide structures. The FT-IR spectrum suggests that the product is likely a polyamide.

Hydrolysis and amino acid analyses were conducted on the ethanol precipitate derived from the cell-free fermentation broth. As depicted in Figure 4c, the analysis of the hydrolysis product and the standard control reveals that the amino acid composition is predominantly glutamic acid (Appendix A), with a retention time of 7 min. Another peak corresponds to the free amino group, observed at 23.40. These findings suggest that the fermentation product by strain SDU is γ-polyglutamic acid.

The molecular weight distribution analysis of γ-PGA produced by SDU is shown in Figure 2d and Appendix A. The weight-average molecular weight (MW) off γ-PGA was 1167 kDa (Appendix A). The molecular weight (MW) ranged from 771 to 3000 kDa, with a polydispersity (MW/Mn) of 1.05 (Appendix A). Under the cultivation conditions used in this study, the molecular weight of γ-PGA produced by *B. velezensis* SDU was mostly above 1000 kDa (Appendix A, Figure 4d). There is no significant variation in the estimated molecular weights between the GPC- and the Mark–Houwink equation-based calculations. The LC-MS analysis of the derivatized products revealed a predominant peak at 10.13 min for the sample, while the retention time ranges for standard D-glutamate and L-glutamate were 10.08–10.19 min (apex at 10.12 min) and 9.83–9.90 min, respectively (Figure 4e).

#### 3.2.4. Preparation of Poly-γ-Glutamic Acid with Different Molecular Weights

The purification process was performed immediately following fermentation, with the viscosity of the sample calculated to be 1390 kDa. The results post-acid degradation are presented in Table 1. It can be observed that the molecular weight of γ-PGA rapidly decreases under acidic conditions, and after approximately 40 min of treatment, its viscosity dropped to a very low level.

#### 3.2.5. Results of ·O_2_^−^ Clearance and Tyrosinase Inhibition by Different Molecular Weights of γ-PGA

As shown in Table 2 and Figure 5, based on the relationship between molecular weight (×10^6^ kDa) and the O_2_^−^ scavenging rate/tyrosinase inhibition rate, a scatter plot was plotted with molecular weight on the horizontal axis and percentage (%) on the vertical axis, incorporating trendlines labeled with their equations and R^2^ values. For the O_2_^−^ scavenging rate versus molecular weight, a strong positive linear correlation (R^2^ > 0.9) was observed, where a 1 × 10^6^ kDa increase in molecular weight corresponded to an approximate 23.61% rise in scavenging rate (Y = 23.61X + 6.50, R^2^ = 0.906). Similarly, the tyrosinase inhibition rate exhibited an extremely strong positive linear correlation with molecular weight (R^2^ > 0.95), showing a 69.06% increase in the inhibition rate per 1 × 10^6^ kDa molecular weight increment (Y = 69.06X − 3.00, R^2^ = 0.953).

## 4. Discussion

*B. subtilis* and *B. licheniformis* are the primary strains for polyglutamic acid production, with key substrates including glutamic acid, sodium glutamate, citrate, fructose, and glucose [27]. Previous studies have also reported the use of glycerol as the main production substrate. After a 4-day fermentation using 80 g/L glycerol and 20 g/L glutamic acid, *B. licheniformis* ATCC9945a produced approximately 17–23 g/L of polyglutamic acid [25]. The conversion rate of polyglutamic acid from the primary nitrogen source, glutamic acid, was 23 g of polyglutamic acid per 80 g glycerol. We identified a strain, *B. velezensis* SDU, capable of utilizing glycerol as the carbon source and ammonium phosphate dibasic taking the place of ammonium chloride as the main nitrogen source. *B. velezensis* SDU can produce up to 23.1 g of polyglutamic acid with only 25 g of glycerol within 30 h. Poly-γ-glutamate (PGA) production is constrained by multiple factors, with dissolved oxygen (DO) identified as a critical limiting factor in the biosynthesis process. Enhanced DO supply could significantly improve yield; however, given the existing high agitation speed approaching operational limits, further optimization should target viscosity reduction through citric acid supplementation to elevate productivity [28]. Concurrently, it must be ensured that the observed viscosity reduction is not attributable to product degradation. We observed the rapid degradation of PGA in the fermenter following nutrient depletion but no corresponding increase in free glutamate levels in the culture medium. This suggests that the degradation mechanism is not end-wise cleavage but rather random-chain scission.

Interestingly, when glycerol was used as the substrate for fermentation, contrary to conventional γ-PGA production media, only D-glutamate was detected as the degradation product of poly-γ-glutamic acid. This observation aligns with previous studies on glutamate-independent amino acid utilization for γ-PGA synthesis [29], but it contrasts with reported findings that γ-PGA produced by glutamate-dependent strains exhibits both D- and L-configurations [30]. The potential correlation between glutamate dependency and the stereochemical composition of γ-PGA warrants further investigation to determine whether this represents a general biological phenomenon. Notably, no systematic studies have been reported in this area to date.

Antioxidant indicators are extensively employed in cosmetics, with the free radical theory speculating a close association between these indicators and the aging process. In one study, a genomic sequence analysis revealed that *Bacillus velezensis*, a polyglutamic acid-producing strain, possesses a unique and distinct mechanism from other Bacillus subtilis strains, potentially endowing its products with unique properties [31]. Tyrosinase, a vital enzyme in melanin formation, serves as a critical benchmark for skincare products aimed at enhancing luminosity. Polyglutamic acid, long recognized as a moisturizer in skincare, demonstrated potential for skin whitening in this study. Poly-γ-glutamic acid solutions with a molecular weight exceeding 1000 KDa demonstrate free radical scavenging and tyrosinase inhibitory activities. In our antioxidant studies, we conducted tyrosinase inhibition tests using polyglutamic acid produced via *B. velezensis* SDU fermentation, revealing that higher-molecular-weight polyglutamic acid exhibits a greater inhibitory rate. Since the concentration of all γ-PGA samples was fixed at 40 g/L, the molar concentration of each molecular weight γ-PGA was calculated at this concentration. For tyrosinase inhibition, the 0.63 × 10^6^ molecular weight sample showed a 52.94% inhibition rate (close to 50%), suggesting its IC_50_ could be approximated as the molar concentration at 40 g/L for this molecular weight (IC_50_ ≈ 6.35 × 10^−5^ M). The observation that none of the molecular weight variants achieved 50% O_2_^−^ scavenging rate at 40 g/L implies that the EC_50_ value of γ-PGA (for superoxide radical scavenging) should exceed 40 g/L under these experimental conditions. γ-PGA underperforms compared to kojic acid and ascorbic acid in direct bioactivity benchmarks. However, its safety, biodegradability, and multifunctionality (e.g., viscosity control and nutrient retention) make it valuable in niche applications where synthetic compounds are less desirable. These findings highlight the importance of molecular weight in determining the efficacy of polyglutamic acid in skincare applications. Despite the many superior properties of polyglutamic acid esters, which stem from their secondary structure, further exploration is necessary. In future studies, attention should also be given to the relationship between molecular weight and its functions.

## 5. Conclusions

This study identified a *B. velezensis* SDU strain that produces polyglutamic acid from glycerol as a substrate. The strain exhibits significantly higher conversion efficiency compared to existing glutamate-independent strains. Even when glutamate-dependent strains are included, although the overall production yield and concentration of polyglutamic acid (PGA) do not demonstrate superiority [32], its high substrate conversion rate nevertheless enables it to serve as a highly efficient production strain, producing 23.1 g/L of polyglutamic acid with just 25 g/L of glycerol within 30 h (Table 3). Using two distinct methods, the average molecular weight of the produced polyglutamic acid consistently exceeded 1000 kDa, confirming that the molecular weight of polyglutamic acid by this strain is high. The scavenging capacity of γ-PGA against superoxide anion (·O_2_^−^) radicals and its inhibitory activity toward tyrosinase were evaluated. The findings indicate that as the molecular weight increases, the polyglutamic acid’s capacity to scavenge free radicals and inhibit tyrosinase also increases exponentially.

## Figures and Tables

**Figure 1 microorganisms-13-00917-f001:**
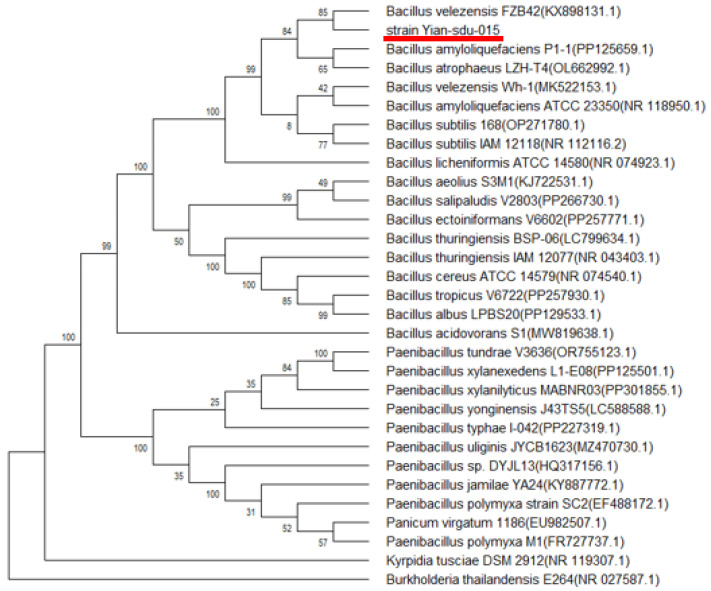
Phylogenetic tree based on 16S rDNA sequences showing the position of *B. velezensis* SDU (red underline) among its closely related organisms. The tree was constructed by the neighbor-joining method. The scale bar represents 0.01 nucleotide substitution per position.

**Figure 2 microorganisms-13-00917-f002:**
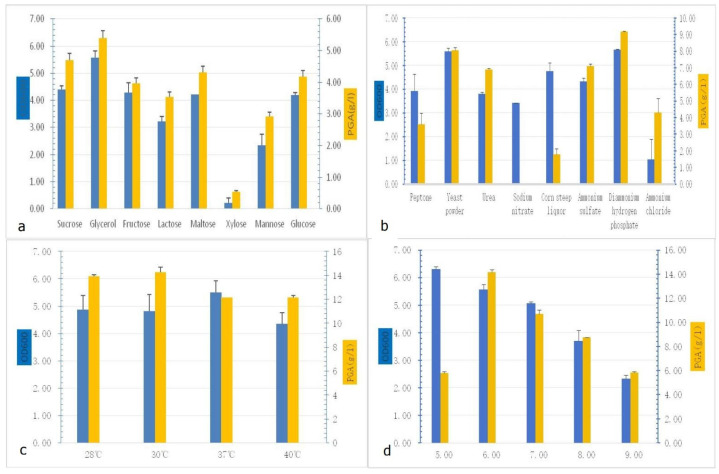
Selection of factors required for γ-PGA production: (**a**) γ-PGA production of different carbon sources; (**b**) γ-PGA production of different nitrogen sources; (**c**) γ-PGA production of different temperatures; (**d**) γ-PGA production of different pH values.

**Figure 3 microorganisms-13-00917-f003:**
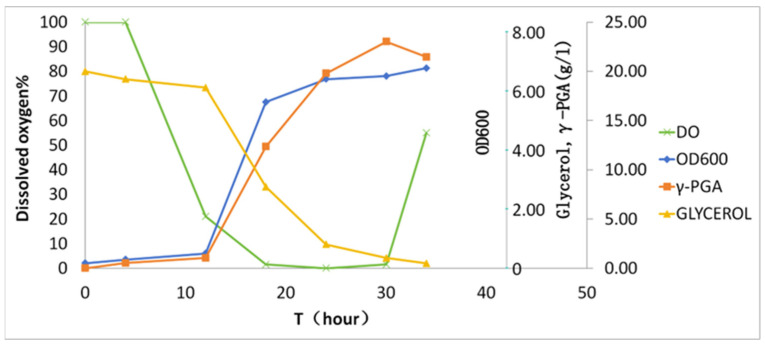
Kinetics of γ-PGA biosynthesis in batch fermentation. The *x*-axis denotes cultivation time; the *y*-axis shares a unified scale for glycerol and γ-PGA given their similar magnitude ranges.

**Figure 4 microorganisms-13-00917-f004:**
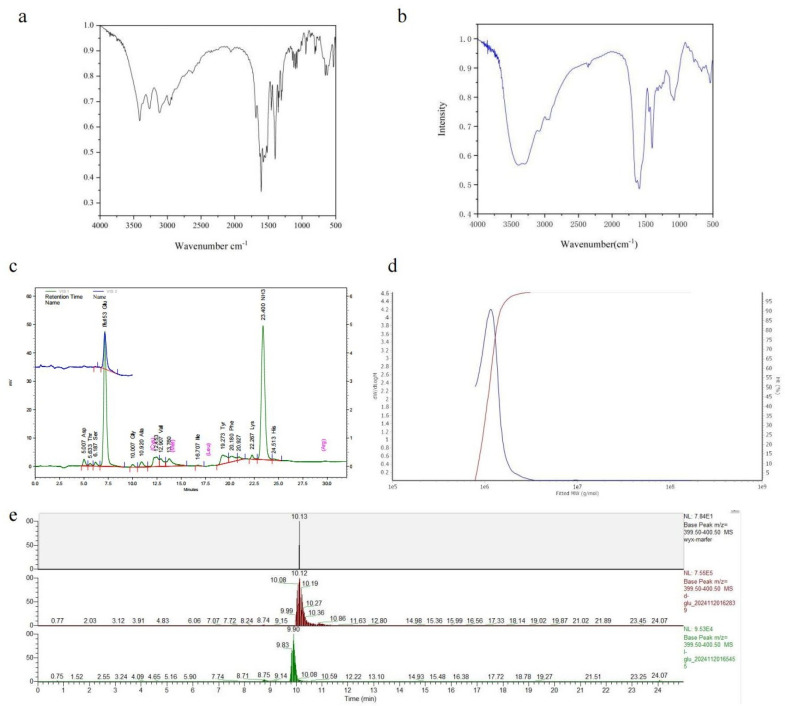
Analysis of the product produced by *B. velezensis* SDU. (**a**) The FT-IR spectra of γ-PGA standard products. (**b**) The FT-IR spectra of the γ-PGA produced by *B. velezensis* SDU. (**c**) Amino acid profiles of the hydrolysis fermentation products of *B. velezensis* SDU. There are mainly two peaks, one for the amino peak retention time of 23 min and the other for the glutamate peak of 7 min. (**d**) GPC-MALLS results of γ-PGA produced by *B. velezensis* SDU. (**e**) LC-MS analysis of the derivatized products.

**Figure 5 microorganisms-13-00917-f005:**
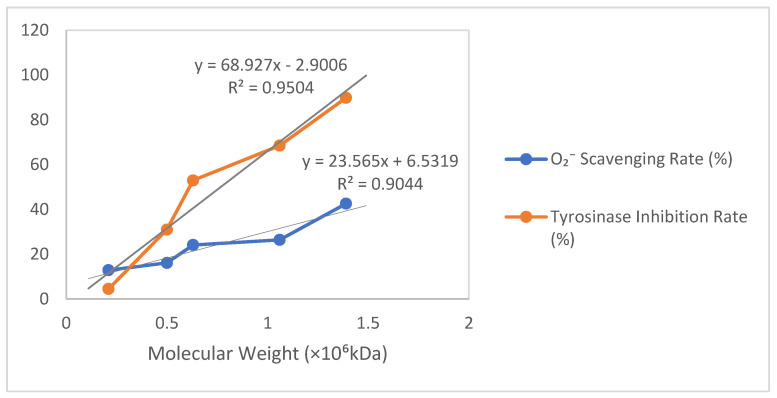
Results of ·O_2_^−^ clearance and tyrosinase inhibition by different molecular weights of γ-PGA produced by *B. velezensis* SDU.

**Table 1 microorganisms-13-00917-t001:** The molecular weights of poly-γ-glutamic acid by different treatments.

Handling Time	0 min	10 min	20 min	30 min	40 min
[η]	24.57	17.95	9.82	7.51	2.75
MW (kDa)	1390	1060	630	500	210

**Table 2 microorganisms-13-00917-t002:** Results of O_2_^−^ clearance and tyrosinase inhibition by different molecular weights of γ-PGA.

Molecular Weight (×10^6^ kDa)	1.39	1.06	0.63	0.50	0.21
O_2_^−^ Scavenging Rate (%)	42.57	26.38	24.08	16.08	12.86
Tyrosinase Inhibition Rate (%)	89.86	68.46	52.94	30.97	4.50

**Table 3 microorganisms-13-00917-t003:** Comparative analysis of glutamate-independent γ-PGA-producing strains.

Isolate	Carbon Source (g·L^−1^)	Methods and Key Conditions	γ-PGA Production (g·L^−1^)	Conversion Efficiency (%) *^a^*	Productivity (g·L^−1^ h^−1^)	References
*B. licheniformis* A35	Glucose (75)	30 °C, 100 h	8.50	16.00	11.33	[33]
*Bacillus subtilis* TAM-4	Glucose (75)	30 °C, 96 h	22.10	29.47	0.23	[34]
*B. amyloliquefaciens* LL3	Sucrose (50)	37 °C, 44 h	4.40	8.80	0.10	[35]
*B. subtilis* GXG-5	Glucose (25)	50 °C, 34 h	19.50	78.00	0.57	[36]
*Bacillus subtilis* C10	Glucose (80) and citric acid (20)	32 °C, 32 h	27.70	27.70	0.79	[37]
*B. licheniformis* TISTR 1010	Glucose (20) and citric acid (30)	37 °C, 95 h	27.50	55.00	0.29	[38]
*B. subtilis* C1	Glycerol (170) and citric acid (22)	37 °C, 6 days	21.40	0.11	0.15	[29]
*Bacillus velezensis* SDU	Glycerol (25) and sodium citrate (3.4)	30 °C, 30 h	23.01	81.00	0.77	This work

*^a^* The conversion efficiency was defined as the ratio of γ-PGA yield to carbon source concentration.

## Data Availability

The datasets generated for this study can be found in online repositories. The names of the repository/repositories and accession number(s) can be found below: https://www.ncbi.nlm.nih.gov/nuccore/PQ066107 (accessed on 29 July 2024).

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
