# Peer review of "Production and Characterization of Poly-γ-Glutamic Acid by Bacillus velezensis SDU"

_microorganisms, 2025, doi:10.3390/microorganisms13040917_

Round 1
Reviewer 1 Report
Comments and Suggestions for Authors
This manuscript presents Bacillus velezensis SDU, a microorganism isolated from the rhizosphere soil of Baimiao taro, as a promising novel strain based on the optimization and characterization of poly gamma-glutamic acid (γ-PGA) production, as well as its antioxidant and enzyme inhibitory activities.
Several revisions and additions are required before the manuscript can be considered for publication:
1.Frequently used expressions should be written in full when first introduced, with abbreviations used for subsequent mentions.
2.Microorganism names should be italicized.
3.The manuscript currently presents chemical formulas such as "NH4Cl" with numerals in regular text. According to standard scientific notation, numerals should be formatted as subscripts (e.g., “NH₄Cl”). Please ensure that all chemical formulas are consistently formatted throughout the manuscript.
4.The overall quality of the figures is low. High-resolution figures should be submitted, the formatting should adhere to the journal’s guidelines, and the figure legends should provide more detailed descriptions.
5.The 16S rRNA analysis shows 99.3% similarity, which is insufficient to claim a novel strain. Additional analyses are required.
6.In the Results section, panel (b) of Figure 3 should be placed after Figure 4.
7.Some parts of section 3.2.5 should be moved to the Discussion section for better organization.
8.The statement that an increase in molecular weight enhances antioxidant effects lacks sufficient supporting evidence. Additional explanation or data is required.
9.The use of "however" in line 361 of the Discussion section is contextually inappropriate and should be revised.
10.The size of the tables should be reduced to improve readability.
11.The final part of the Introduction lacks a clear explanation of the study's objectives. Additionally, the relationship between previously mentioned studies on γ-PGA molecular weight and the current research should be more clearly articulated.
Please address these issues to improve the completeness and clarity of the manuscript.
Author Response
|
1. Summary |
|
|
Thank you very much for taking the time to review this manuscript. Please find the detailed responses below and the corresponding revisions/corrections highlighted/in track changes in the re-submitted files.
|
|
|
3. Point-by-point response to Comments and Suggestions for Authors Comments 1: Frequently used expressions should be written in full when first introduced, with abbreviations used for subsequent mentions. |
|
|
Response 1: Thank you for pointing this out. We agree with this comment. Therefore, We have included parenthetical explanations when first using the poly-γ-glutamate (γ-PGA) abbreviation in lines 18-19 of the text. |
|
|
Comments 2: [Microorganism names should be italicized.] |
|
|
Response 2: Agree. We have modified to emphasize this point. Revisions include Bacillus subtilis and Escherichia coli in line 51, and Escherichia coli's position in line 110.
Comments 3: [The manuscript currently presents chemical formulas such as "NH4Cl" with numerals in regular text. According to standard scientific notation, numerals should be formatted as subscripts (e.g., “NH₄Cl”). Please ensure that all chemical formulas are consistently formatted throughout the manuscript.] Response 3: Thanks for your attention and questions. We have modified Corrections include MnSO4·H2O (line 109) and (NH4)2SO4, (NH4)2HPO4, NaNO3 (lines 130-131).
Comments 4: [The overall quality of the figures is low. High-resolution figures should be submitted, the formatting should adhere to the journal’s guidelines, and the figure legends should provide more detailed descriptions] Response 4: Thanks for your attention and questions. We have improved the resolution of Figure 2 (as noted in lines 281-282 of the text) and supplemented the axis label explanations for Figure 3 in lines 306-308.
Comments 5: The 16S rRNA analysis shows 99.3% similarity, which is insufficient to claim a novel strain. Additional analyses are required. Response 5: Agree,Since Bacillus velezensis is not a novel species but a newly identified producer of poly-γ-glutamic acid (γ-PGA), line 18 has been revised to: "A Bacillus velezensis SDU strain capable of producing poly-γ-glutamate (γ-PGA) was newly identified from the rhizosphere soil of Baimiao taro." Comments 6: In the Results section, panel (b) of Figure 3 should be placed after Figure 4. Response 6: Agree,Figure captions (lines 300-305) were moved to line 375 to improve clarity, as the original merged formatting reduced legibility.
Comments 7:Some parts of section 3.2.5 should be moved to the Discussion section for better organization. Response 7: Agree,The content in lines 356-365 of the manuscript has been strictly restricted to the presentation and analysis of results and data, while the discussion section has been relocated to lines 413-423.
Comments 8:The statement that an increase in molecular weight enhances antioxidant effects lacks sufficient supporting evidence. Additional explanation or data is required. Response 8: Agree,The rationale for formulating this hypothesis is explicitly addressed in the Results and Discussion section, supported by experimental data analysis to lines 413-423.
Comments 9:The use of "however" in line 361 of the Discussion section is contextually inappropriate and should be revised. Response 9: Thanks for your attention and questions. We have opted to use the conjunction "but"(line397) to enhance contextual coherence.
Comments 10:The size of the tables should be reduced to improve readability. Response 10:Agree,the tables in lines 353, 367, and 443 of the manuscript have been adjusted to align properly with the surrounding text for improved contextual coherence.
Comments 11:The final part of the Introduction lacks a clear explanation of the study's objectives. Additionally, the relationship between previously mentioned studies on γ-PGA molecular weight and the current research should be more clearly articulated. Response 11:Thanks for your attention and questions. In lines 83-92 of the manuscript, we have provided a comprehensive account of the rationale and methodological necessity underpinning this study. |
|
Reviewer 2 Report
Comments and Suggestions for Authors
This study presents a newly isolated strain of Bacillus velezensis (SDU), derived from rhizosphere soil, which emerges as a highly effective producer of high molecular weight poly-γ-glutamic acid (γ-PGA) under conditions that do not require exogenous glutamate.
Areas for Improvement:
The isolation steps are well-articulated, following established microbiological protocols. Identification through 16S rDNA sequencing and phylogenetic analysis provides a robust taxonomic classification within the Bacillus velezensis group. The Authors can incorporate a biochemical or phenotypic profile table, which could offer additional insight into the strain's metabolic capabilities—particularly useful for industrial scalability.
A one-variable-at-a-time (OVAT) approach was used to fine-tune medium composition and culture parameters, with glycerol and ammonium phosphate emerging as optimal. The experimental design included triplicates, ensuring data validity. Complementing OVAT with a statistical design, such as response surface methodology, may uncover variable interactions and yield further enhancements.
Production was successfully scaled in a 50-L bioreactor, with continuous monitoring of pH, dissolved oxygen, and agitation. Notably, the use of DO rebound as a termination signal adds practicality to the process. Deeper analysis must be included.
The downstream purification workflow is sound, and the polymer’s identity was rigorously confirmed via FT-IR, amino acid analysis, and LC-MS after Marfey’s derivatization to verify the D-glutamate configuration. Molecular weight was determined through both the Mark-Houwink relationship and GPC-MALLS.
The paper successfully connects γ-PGA’s bioactivity—particularly superoxide radical scavenging and tyrosinase inhibition—with its molecular weight. These functions are of direct relevance to cosmetic and therapeutic applications.To strengthen these findings:
- Report IC50 values for both assays.
- Benchmark γ-PGA’s performance against established compounds (e.g., kojic acid, ascorbic acid).
- Investigate additional biofunctional properties, such as metal ion binding, cryoprotection, or film-forming behavior.
Discuss whether degradation occurs through end-chain cleavage or random hydrolysis, as this distinction can influence downstream properties like solubility and viscosity.
Employ tests such as ANOVA or t-tests to assess significance between production variables and bioactivity levels. Including confidence intervals would also enhance the robustness of the data.
The performance summary table (Table 3) offers a helpful overview of how this strain compares to others, positioning it within the broader research context.
Author Response
Thank you very much for taking the time to review this manuscript. Please find the detailed responses below and the corresponding revisions/corrections highlighted/in track changes in the re-submitted files.
3. Point-by-point response to Comments and Suggestions for Authors
Comments 1: The isolation steps are well-articulated, following established microbiological protocols. Identification through 16S rDNA sequencing and phylogenetic analysis provides a robust taxonomic classification within the Bacillus velezensis group. The Authors can incorporate a biochemical or phenotypic profile table, which could offer additional insight into the strain's metabolic capabilities—particularly useful for industrial scalability.
Response 1: Thank you for pointing this out. We agree with this comment. Therefore, descriptions of colony morphological characteristics have been supplemented in lines 246–249 of the manuscript.
Comments 2: A one-variable-at-a-time (OVAT) approach was used to fine-tune medium composition and culture parameters, with glycerol and ammonium phosphate emerging as optimal. The experimental design included triplicates, ensuring data validity. Complementing OVAT with a statistical design, such as response surface methodology, may uncover variable interactions and yield further enhancements.
Response 2: Thanks for your attention and questions. In lines 279-282, we have explained the omission of the one-variable-at-a-time (OVAT) experimental approach. Following single-factor shake-flask optimization, the observed high viscosity attributed to the elevated molecular weight of γ-PGA necessitated a direct transition to scale-up fermentation in bioreactors.
Comments 3: Production was successfully scaled in a 50-L bioreactor, with continuous monitoring of pH, dissolved oxygen, and agitation. Notably, the use of DO rebound as a termination signal adds practicality to the process. Deeper analysis must be included.
Response 3: The observed dissolved oxygen (DO) rebound phenomenon is addressed in two key sections: Line 298 now specifies that DO recovery occurred concurrent with glycerol depletion, suggesting substrate exhaustion-triggered metabolic shifts. Furthermore, lines 382–392systematically highlight DO as the critical limiting factor in γ-PGA biosynthesis.
Comments 4: The paper successfully connects γ-PGA’s bioactivity—particularly superoxide radical scavenging and tyrosinase inhibition—with its molecular weight. These functions are of direct relevance to cosmetic and therapeutic applications. To strengthen these findings:
•Report IC50 values for both assays.
•Benchmark γ-PGA’s performance against established compounds (e.g., kojic acid, ascorbic acid).(414-424)
•Investigate additional biofunctional properties, such as metal ion binding, cryoprotection, or film-forming behavior.
Response 4: Thanks for your attention and questions. In lines 414-424 of the manuscript, we discuss the half-maximal inhibitory concentration (IC₅₀) values obtained from both bioactivity assays. Additionally, the performance of γ-PGA is benchmarked against established bioactive compounds (e.g., kojic acid and ascorbic acid). In lines 83–92 of the manuscript, we have revised the rationale for assessing the antioxidant and tyrosinase inhibitory activities of γ-PGA, specifically to systematically evaluate how controlled structural degradation impacts its bioactive efficacy. This experimental design directly addresses the stability-performance relationship critical for real-world applications in functional cosmetics and biomedical formulations.
Comments 5: Discuss whether degradation occurs through end-chain cleavage or random hydrolysis, as this distinction can influence downstream properties like solubility and viscosity.
Response 5: Thanks for your attention and questions. The degradation mechanisms of γ-PGA are systematically analyzed in lines 390–393
Comments 6: Employ tests such as ANOVA or t-tests to assess significance between production variables and bioactivity levels. Including confidence intervals would also enhance the robustness of the data.
Response 6: Agree. In the antioxidant activity analysis (lines 357–371), we have incorporated trendline regression analysis to statistically validate the dose-dependent relationship between γ-PGA concentration and radical scavenging capacity, with residual plots confirming model robustness.
Comments 7: The performance summary table (Table 3) offers a helpful overview of how this strain compares to others, positioning it within the broader research context.
Response 7: We have introduced additional discussion in lines 431–435 of the manuscript and broadened the scope to encompass a wider range of γ-PGA-producing microbial strains.
Reviewer 3 Report
Comments and Suggestions for Authors
Production and Characterization of Poly-γ-Glutamic Acid by Bacillus velezensis SDU 3
by Guangyao Guo 1,2,3,#, Han Wang 2,#, Huiyuan Jia 2,4, Haiping Ni 2,+, Shouying Xu 2, Cuiying Zhang 1, Youming Zhang 4 2,3,5*, Yuxia Wu 2 and Qiang Tu 2,3,5*
Research on Poly-γ-Glutamic Acid is intense at a worldwide level
With many many applications
Bacillus strains are screened
Poly-γ-Glutamic Acid from a Novel Bacillus subtilis Strain: Strengthening the Skin Barrier and Improving Moisture Retention in Keratinocytes and a Reconstructed Skin Model
Ko, H.-J., Park, S., Shin, E., ... Lee, J., Hyun, C.-G.
International Journal of Molecular Sciences, 2025, 26(3), 983
however it is difficult to understand if these ‘novel’ strains can compete with Engineered B. subtilis strains (example below)
Citrate Supplementation Modulates Medium Viscosity and Poly-γ-Glutamic Acid Synthesis by Engineered B. subtilis 168
Völker, F., Hoffmann, K., Halmschlag, B., ... Büchs, J., Blank, L.M.
Engineering in Life Sciences, 2025, 25(3), e70009
______________________________
The γ-PGA production was reached 23.1 g/L and the productivity was 0.77g L−1 h−1.
could authors provide a table with comparison of their results (g/L, g L−1 h−1) with other results from 2020-2025?
some data are already in Table 3
----------------------------------------
Are there already some industrial production of Poly-γ-Glutamic Acid?
companies, countries, trademarks
price per ton
---------------------------
Since the discovery of polyglutamic acid in the last century, extensive research has been conducted on its fermentation level and scaleup process, with a fermentation yield of up to 41.40 ± 2.01 g/L in the repeated fed-batch fermentation, and new strains and production processes have been continuously discovered
fully true, that is why I asked the previous question
______________________________
Poly-(γ-glutamic acid) production and optimization from agro-industrial bioresources as renewable substrates by bacillus sp. FBL-2 through response surface methodology
Song, D.-Y., Reddy, L.V., Charalampopoulos, D., Wee, Y.-J.
Biomolecules, 2019, 9(12), 754
usual C and N substrates OR agro-industrial bioresources?
what do you recommend?
---------------
Author Response
1. Summary
Thank you very much for taking the time to review this manuscript. Please find the detailed responses below and the corresponding revisions/corrections highlighted/in track changes in the re-submitted files.
3. Point-by-point response to Comments and Suggestions for Authors
Comments 1: Research on Poly-γ-Glutamic Acid is intense at a worldwide level
With many many applications Bacillus strains are screened
Poly-γ-Glutamic Acid from a Novel Bacillus subtilis Strain: Strengthening the Skin Barrier and Improving Moisture Retention in Keratinocytes and a Reconstructed Skin Model
Ko, H.-J., Park, S., Shin, E., ... Lee, J., Hyun, C.-G.
International Journal of Molecular Sciences, 2025, 26(3), 983
however it is difficult to understand if these ‘novel’ strains can compete with Engineered B. subtilis strains (example below)
Citrate Supplementation Modulates Medium Viscosity and Poly-γ-Glutamic Acid Synthesis by Engineered B. subtilis 168
Völker, F., Hoffmann, K., Halmschlag, B., ... Büchs, J., Blank, L.M.
Engineering in Life Sciences, 2025, 25(3), e70009
Response 1: Thank you for pointing this out. We agree with this comment. Therefore we have provided clarifications in lines 78-82 of the manuscript, and the factors influencing poly-γ-glutamic acid (γ-PGA) yield are systematically discussed in line 382-392.
Comments 2: could authors provide a table with comparison of their results (g/L, g L−1 h−1) with other results from 2020-2025?
Response 2: Thanks for your attention and questions. In lines 424-429 of the manuscript, we analyze the production efficiency of microbial strains. This discussion is supplemented by a 2017 reference that catalogs glutamate-dependent γ-PGA-producing strains, highlighting their historical production capabilities. Notably, while high-yield γ-PGA production prior to 2020 has been well-documented in glutamate-dependent systems, reports of highly efficient glutamate-independent producers within the past five years remain scarce.
Comments 3: Are there already some industrial production of Poly-γ-Glutamic Acid? companies, countries, trademarks price per ton
Response 3: Thanks for your attention and questions. As outlined in lines 79-82 of the manuscript, despite the commercial availability of certain products, their widespread application remains constrained by prohibitively high costs and challenges in macromolecular stability. Identifying optimal application scenarios could significantly enhance the competitiveness and value proposition of these materials.
Comments 4: [Poly-(γ-glutamic acid) production and optimization from agro-industrial bioresources as renewable substrates by bacillus sp. FBL-2 through response surface methodology
Song, D.-Y., Reddy, L.V., Charalampopoulos, D., Wee, Y.-J.
Biomolecules, 2019, 9(12), 754
usual C and N substrates OR agro-industrial bioresources?
what do you recommend?
Response 4: Thanks for your attention and questions. The stereochemical configuration (D/L ratio) of γ-PGA produced by glutamate-independent or dependent strains may vary depending on strain specificity and cultivation media, thereby influencing functional properties. However, the relationship between media composition and γ-PGA structural characteristics remains underexplored. As highlighted in lines 79-82, the relatively high production costs of γ-PGA necessitate broader media selectivity optimization to enhance cost-effectiveness for industrial applications.
Round 2
Reviewer 2 Report
Comments and Suggestions for Authors/